# Socio-Epidemiological Features and Spatial Distribution of Malaria in an Area under Mining Activity in the Brazilian Amazon Region

**DOI:** 10.3390/ijerph181910384

**Published:** 2021-10-02

**Authors:** Thalyta Mariany Rêgo Lopes Ueno, Luana Nepomuceno Gondim Costa Lima, Daniele Melo Sardinha, Yan Corrêa Rodrigues, Herberto Ueno Seelig de Souza, Paula Ribeiro Teixeira, Ricardo José de Paula Souza e Guimarães, Karla Valéria Batista Lima, Ana Maria Revorêdo da Silva Ventura

**Affiliations:** 1Programa de Pós-Graduação em Biologia Parasitária na Amazônia (PPGBPA), Instituto Evandro Chagas (IEC), Universidade do Estado do Pará (UEPA), Belém 66087-662, Brazil; luanalima@iec.gov.br (L.N.G.C.L.); yan.13@hotmail.com (Y.C.R.); karlalima@iec.gov.br (K.V.B.L.); ana_mariaventura@hotmail.com (A.M.R.d.S.V.); 2Seção de Bacteriologia e Micologia (SABMI), Instituto Evandro Chagas (IEC), Ananindeua 67030-000, Brazil; danielle-vianna20@hotmail.com; 3Programa de Pós-Graduação em Epidemiologia e Vigilância em Saúde (PPGEVS), Instituto Evandro Chagas (IEC), Ananindeua 67030-000, Brazil; ricardoguimaraes@iec.gov.br; 4Programa de Pós-Graduação em Ciências Florestais (PPGCF), Universidade Federal Rural da Amazônia (UFRA), Belém 66077-830, Brazil; uenoambiental@gmail.com; 5Discente do Curso de Licenciatura em Ciências Biológicas, Instituto de Educação, Ciência e Tecnologia do, Pará (IFPA), Itaituba 68183-300, Brazil; paula.tecminer@gmail.com; 6Laboratório de Geoprocessamento (LABGEO), Instituto Evandro Chagas (IEC), Ananindeua 67030-000, Brazil; 7Seção de Parasitologia (SAPAR), Instituto Evandro Chagas (IEC), Ananindeua 67030-000, Brazil

**Keywords:** Amazonia, malaria, mining, spatial analysis

## Abstract

Malaria is an acute febrile infectious disease that represents an important public health problem in the Brazilian amazon region. The present study described the socio-epidemiological and spatial characteristics of malaria in a population from the Tapajós mining areas, Pará, Brazilian Amazon. A cross-sectional study, including individuals from Itaituba city, an area under mining activity influence, was conducted. The geographic coordinates were obtained in the field using the Global Positioning System (GPS) Garmin 78csx; for spatial analysis, we used the Kernel Density Estimator with the application of scanning statistics with the SaTScan software. Of the 908 individuals, 311 were positive for malaria. Most of the malaria cases were associated with male individuals, gold miners and with a monthly income of 4-6 salaries. Binary logistic regression analysis demonstrated that gold miners were nearly five times more likely to acquire malaria. In addition, a context of risk for sexually transmitted infections, substance abuse and poor support conditions was observed, worsening the healthcare scenario in this endemic area for malaria. The spatial distribution of malaria cases is irregular in the municipality with hotspot areas located in the Amana Flona that coincide with areas of illegal mining and high human mobility. Finally, the presented socio-epidemiological and spatial distribution data may aid in the development of more effective control measures for malaria in the area.

## 1. Introduction

Malaria is an acute febrile infectious disease, which stands as a major endemic public health issue in tropical and developing regions. This vector-borne parasitic disease is caused by six species of *Plasmodium*, including *P. falciparum, P. vivax, P. malariae, P. ovale wallickeri, P. ovale curtisi and P. knowlesi.* In Brazil, malaria cases are caused only by the first three cited species, which are predominantly transmitted by *Anopheles darlingi* mosquitoes [1,2].

In 2018, worldwide, approximately 228 million cases of malaria were registered, of which 93% were in the African Region, followed by Southeast Asia (3.4%) and the eastern Mediterranean regions (2.1%). In Latin America and the Caribbean regions, malaria is endemic in 21 countries, with 120 million people at risk of infection. In South America, nine countries, including Brazil, currently account for 93.2% of malaria cases on the continent. Although several characteristics are associated with malaria cases in different countries, all the presented regions still have limited and/or lack of access to malaria prevention and treatment services for the population [3,4].

The majority of malaria cases in Brazil occur in the Amazon region. Within the region, in 2019, the State of Pará was the second in the incidence of cases, even with an expressive 29.4% reduction compared to 2018. The city of Itaituba, located near the Tapajós River, in Pará state, showed a 42.1% increase in malaria cases (2560/3651) [5]. Itaituba city presents favorable hydrographic, climatic and environmental conditions, which enables the interaction between the human host and the vectors, favoring the transmission and maintenance of the malaria cycle. A remarkable feature observed in the proximities of the city is the existence of mining activity, which contributes to high population mobility, due to the constant presence of individuals looking for work, causing a disorderly growth and the poor development of socioeconomic conditions in the region.

Additionally, the vulnerability context of this population enhances the number of other disease cases, including sexually transmitted infections (STIs), Chagas disease, dengue fever and others endemic diseases [6,7]. In this sense, historically, the occurrence of co-infection by malaria and other diseases is often observed in these areas, contributing to the high level of endemicity and chances of re-infection episodes during their lifetime [2,8,9].

As previously presented, effective malaria control strategies need to recognize epidemiological particularities and establish measures according to the characteristics of each location [8,10]. The present study describes the socio-epidemiological and spatial features associated with malaria in a mining activity setting in the State of Para, Brazilian Amazon region.

## 2. Materials and Methods

### 2.1. Study Delineation

From March to December 2019, a cross-sectional study, including individuals from Itaituba city, a location under mining activity in the Brazilian Amazon region, was conducted. According to 2019 data, the city of Itaituba has an estimated population of 101,247 people, in an area of 62,042.472 km^2^, with a population density of 1.57 hab/km^2^; human development index (HDI) of 0.640, which is below the Brazilian average (0.727); and its gross domestic product (GDP) per capita is 17,971.96 [11].

### 2.2. Inclusion Criteria and Ethical Considerations

Individuals over 10 years of age, who sought healthcare assistance at the Endemic diseases sector in the Municipality Hospital of Itatituba, presenting suggestive symptoms of malaria and/or positive testing in thick blood smears, composed the evaluated population. Individuals who agreed to participate in the study were interviewed, answering a structured questionnaire for socio-epidemiological data collection. After the interview, 5 mL of blood was collected by venipuncture for the thick blood smears for malaria testing, and for the rapid test and Venereal Disease Research Laboratory (VDRL) for syphilis. Individuals who were unable (or refused) to answer the questionnaire, and/or refused to provide blood samples, were excluded from the study.

Written informed consent was obtained from all enrolled individuals. This study was approved by the Research Ethics Committee of the Pará State University, under number 2,852,618, on 29 August 2018.

### 2.3. Statistical Analysis

For the comparison analysis of socio-epidemiological variables, the chi-square test of adherence and chi-square test of independence were performed when applicable. The G-test of independence was applied to verify the association between variables among positive and negative cases of malaria. Binary logistic regression was performed with the dependent variable positive for malaria, calculating odds ratios (ORs) and the 95% confidence intervals (CIs). The software the Statistical Package for the Social Sciences 20.0 was used for statistical analysis and *p*-values ≤ 0.05 were considered statistically significant.

### 2.4. Spatial Distribution Analysis

State and municipal boundaries and census sectors [11] were obtained from the Brazilian Institute of Geography and Statistics (IBGE), and the geographic coordinates of the resident localities of the participants were obtained with a Garmin 78csx Global Positioning System (GPS) and used to create a geographic database (BDG). The BDG was imported into a geographic information system (GIS) where the following analyses were performed: spatial distribution of malaria cases and identification of risk areas.

For the spatial pattern of malaria dynamics considering the geographical coordinates of localities with the transmission, the Kernel Density Estimator (EDK) was used as an analytical method. The EDK is an estimator that produces a density surface of points per unit area [12,13]. TerraView program (http://www.dpi.inpe.br/terralib5/wiki/doku.php, accessed on 4 May 2021) was used, the parameters of EDK were karmic function, density calculation and adaptive radius. Scan statistics (Scan) was also applied, using SaTScan software (https://www.satscan.org/, accessed on 4 May 2021). SaTScan analyzes spatial, temporal and Spatio-temporal data to detect spatial or Spatio-temporal disease clusters and to see if they are statistically significant [14]. Bernoulli’s model was used to determine the risk area using malaria case and control data.

## 3. Results

A total of 908 individuals participated in the study. The socio-epidemiological data demonstrated that most of the participants were male (58.1%-528/908), within the age range of 21 to 30 years (31.8%-289/908), self-declared brown (75.9%-689/908) and employed as gold miner (gold digger) (42.8%-389/908) (Table 1). Among the positive cases for malaria (34.2%-311/908), 67.8% (211/311) were male, aged between 21 and 30 years (37.3%-116/311) and workers at mining sites within the region (61.7%-192/311). Malaria cases were significantly predominant among male and gold miners, as presented in Table 1 (*p* < 0.001). The binary logistic regression model (*p* < 0.001) demonstrated that working as a gold miner (OR 4.387, CI 2.744–7.012/*p* < 0.001) and with an income in the range of 2–3 salaries (OR 1.928, CI 1.355–2.745/*p* < 0.001) increased the chances of acquiring malaria by nearly five and two times, respectively (Table 2).

Regarding the etiological agent, *P. vivax* was associated with the majority of malaria cases (90.4%-281/311) (*p* < 0.001), followed by *P. falciparum* (7.7%-24/311) and mixed infection by *P. vivax* and *P. falciparum* (1.9%-6/311). Additionally, 63.3% (197/311) of the participants had a qualitative parasitemia of 2+ and quantitative parasitemia between 501 and 10,000 parasites per mm3 (Table 3).

Among the individuals positive for malaria, 13.1% (41/311) had a positive result on the rapid test for syphilis and 9.3% (29/311) on the VDRL essay. Only 8.0% (25/311) reported having an STI in the last 12 months, of which 44% (11/25) did not receive any treatment. Approximately 47.0% (146/311) had been tested for HIV at least once, and none confirmed seropositivity (Table 4).

The use of drugs in the last 12 months was reported by 78.1% (243/311) of the participants, of which, 94.6% (230/243) mentioning alcohol and marijuana (13.6%-33/243) as the most often used licit and illicit drugs, respectively (Table 3). Regarding the type of partner, 66.9% (208/311) reported having sexual relations with women, and 52.7% (164/311) with only one partner in the last thirty days. Gender identity and sexual orientation information revealed that 68.49% (213/311) declared themselves as male, and heterosexual (95.8%-298/311) (Table 4).

Figure 1 shows the spatial distribution of malaria cases (positive in red) and controls (negative—green) in the city of Itaituba, Pará, Brazil. Most of the positive cases had Flona Amana (37.3%-116/311) as the probable infection site, followed by Itaituba Zone-2 (Garimpos of Transgarimpeira) (15.7%-49/311) and Itaituba Zone (Garimpos of Rio Bom Jardim) (14.8%-46/311).

The EDK and SaTScan applications revealed the presence of one high-risk cluster and three other low-risk clusters for malaria transmission in the city of Itaituba, Pará. The high-risk localities were Flona Amana (Pista, Salto, CVA, Garimpinho and Cara Preta) and Porquinho. SaTScan identified a significant cluster (*p*-value = 0.00014) with a radius of 214.6 km and a relative risk of 1.32 at the same location as the high-risk cluster identified using the EDK (Figure 2).

## 4. Discussion

Gold mining plays a major role in malaria maintenance worldwide. In the Americas and Brazil, most of the malaria cases are registered in the Amazon region, with transmission restricted to rural hotspot areas, mainly with the presence of gold mining activity; in the states of Acre and Amazonas; and the cities of Porto Velho (State of Rondônia), Itaituba and Anajás (State of Pará) [8,15,16]. In 2019, Pará state was the second in the number of malaria cases (32,752 cases) with 75.5% occurring in rural areas, 13% in mining areas, 8.1% in indigenous areas and only 2.7% in urban areas [5]. The present study investigated the spatial and epidemiological context of malaria in a mining activity setting, and one of the most important areas for malaria endemicity in the country.

The data from the Brazilian Ministry of Health assures that 41 municipalities in the Amazon region account for over 80% of the notifications of malaria cases, which are predominantly caused by *P. vivax*, and in parallel with a decreasing incidence of *P. falciparum* over the years. The extra-Amazon region malaria cases account for only 1% of the notified cases in the country [5,17,18,19].

The endemicity for malaria in the Amazon rainforest is strongly related to environmental factors, such as rain, temperature, air humidity and water sources, which are fundamental in maintaining the vector cycle and disease transmission. In addition, the sociocultural (socio-environmental) dynamics also play a fundamental role in the complexity of malaria in the region [20,21]. As an example, the presence of streams and small rivers derived from Tapajós river, and small dams built by miners for water accumulation and consumption provide an environment for the development of *Plasmodium* vectors (*Anopheles* mosquitoes). Regarding the use of prophylactic measures, the use of mosquito nets is common in this population, while the intake of antimalarials is rarely observed.

In line with previous reports, our data demonstrate that the gold mining areas near Itaituba city contribute significantly to the high prevalence of malaria, especially among men, miners, within the age range of 21–30 years. Additionally, this population is highly vulnerable due to an unfavorable socio-economic context, including lack or limited access to health services, poor living conditions and abuse of licit and illicit drugs, which enhances the spread other infectious diseases and worsens the healthcare scenario in the region [7,9,16,22]. In spite of that, behavioral data on individuals with malaria in mining areas are scarce in the literature, thus, our data involving these aspects may contribute to a better understanding of this context.

Furthermore, Itaituba is the main city in the southwest region of Pará, leading to the continuous and long-term migration of individuals from neighboring towns and other locations who are in search of work (mainly, ‘enrichment’ opportunities in mining activities) and treatment for malaria and other diseases, also favoring the movement of goods and services, among which is mining. Unfortunately, due to a lack of proper government strategies and income alternatives, the economic factor motivates the return of individuals to these spaces, even at the known risk of contracting malaria and other conditions [16,23,24].

In a study conducted in Boa Vista, Roraima, Brazil, Louzada et al. [16] highlight the migration flow as an important factor for the maintenance of malaria endemicity in the region, similarly to that observed in the city of Itaituba. Finally, in this context, human migration increases the risk of contracting malaria by about thirty times [25].

In the present study, spatial epidemiology contributed to the identification of hotspots with the support of EDK and SatScan technology, allowing targeted interventions. Among the 311 cases of malaria detected in the present study, 37.3% (116/311) of individuals are likely to have been infected in the Amana National Forest (Flona), an Environmental Conservation Unit, with approximately 542,000 inhabitants, located in the municipalities of Itaituba and Jacareacanga-PA, on the border with the State of Amazonas. Within the cluster identified in the Amana Flona, the following sites were classified as at highest risk of malaria transmission: Pista, Salto, CVA, Garimpinho and Cara Preta and Porquinho, sites with heavily illegal mining activity according to the National System of Conservation Units (SNUC) [26,27]. Several issues need solutions and attention from health, sanitary and environmental control agencies, not only because of the threat to the fauna and flora, but also due to the large population flow in this area [26]. In Itaituba, the cases of malaria were related to the mining activity, evidencing that the advance of man over nature generates unbalance and, in this sense, control measures for malaria should include early diagnosis and treatment of malaria cases (through active and passive searching); monitoring of endemic sites for malaria, in parallel with control of the migratory flow; and health education involving not only the miners but also other individuals.

Recently, Lopes et al. [6] highlighted the importance of sharing study results with local and state managers, which contributes to a better distribution of resources and planning of specific actions in the areas most affected by malaria. The disease control measures must be in line with other spheres of public power, for example in the offer of jobs that are not related to man’s need to enter the environment and degrade forest areas for subsistence.

The incidence of malaria is influenced by factors related to regional characteristics; therefore, appropriate disease and vector control strategies should be implemented in each locality [28]. We emphasize the need for further studies in the southwestern region of Pará state, since this territorial extension presents numerous areas of gold and mineral extraction; thus, it would be possible to evaluate the context and identify other clusters at risk of malaria transmission, as well as helping to better control and eliminate malaria in an area of the Amazon rainforest.

## 5. Conclusions

The present study demonstrates that malaria is still a major problem in Itaituba, Pará, with irregular distribution of the disease, along with high-risk transmission areas in illegal mining sites, mainly in the conservation unit of Flona Amana. The migratory flow of individuals who work in mining activity also contributes to the maintenance of endemicity and risk of acquiring malaria. Finally, we highlight that the social-epidemiological context of poor health conditions, transmission of STIs and abuse of licit and illicit drugs poorly reflects on the dynamics of malaria control.

## Figures and Tables

**Figure 1 ijerph-18-10384-f001:**
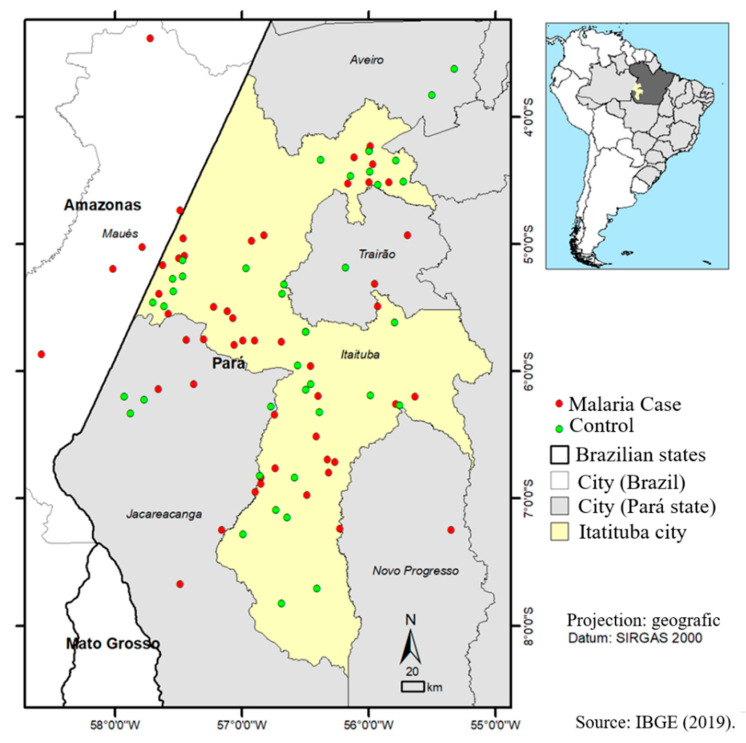
Distribution of malaria cases and control cases in the municipality of Itaituba-Pará.

**Figure 2 ijerph-18-10384-f002:**
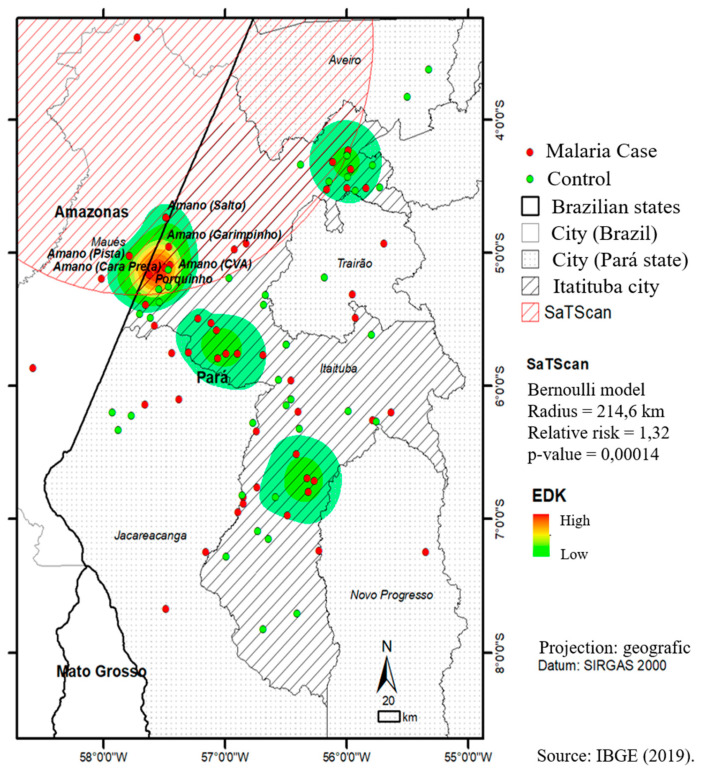
High and low risk clusters for malaria in the city of Itaituba, Pará, Brazilian Amazon.

**Table 1 ijerph-18-10384-t001:** Socio-demographic characteristics of individuals from Itatituba city, Pará state, Brazilian Amazon.

Variables	N	%	*p*-Value *	Positive Malaria	%	Negative Malaria	%	*p*-Value ** Positive vs. Negative
**Sex**								
Male	528	58.15	<0.001	211	67.85	317	53.10	<0.001
Female	380	41.85		100	32.15	280	46.90	
Age group								
<=20	81	8.92	<0.001	33	10.61	48	8.04	0.432
21–30	289	31.83		116	37.30	173	28.98	0.132
31–40	241	26.54		78	25.08	163	27.30	0.217
41–50	156	17.18		53	17.04	103	17.25	0.899
>=51	141	15.53		31	9.97	110	18.43	0.012
Marital status								
Married	535	58.92	<0.001	165	53.05	370	61.98	0.117
Separate	51	5.62		15	4.82	36	6.03	0.500
Single	310	34.14		130	41.80	180	30.15	0.006
Widower	12	1.32		1	0.32	11	1.84	0.827
Race								
Brown	689	75.89		243	78.14	446	74.71	0.873
Black	113	12.44	<0.001	35	11.25	78	13.07	0.973
White	105	11.56		33	10.61	72	12.06	0.901
Indigenous	1	0.11		0	0	1	0.17	0.464
Salary ^#^ range								
<1 salary	114	12.56		21	6.75	93	15.58	<0.002
1–3 salaries	344	37.89	<0.001	116	37.30	228	38.19	0.487
4–6 salaries	70	7.71		62	19.94	8	1.34	<0.001
>6 salaries	47	5.18		25	8.04	22	3.69	0.008
uninformed	333	36.67		87	27.97	246	41.21	<0.001
Schooling								
Illiterate	23	2.53		6	1.93	17	2.85	0.397
ES incomplete	469	51.65		164	52.73	305	51.09	0.887
ES Complete	86	9.47	<0.001	35	11.25	51	8.54	0.283
HS Incomplete	123	13.55		45	14.47	78	13.07	0.280
HS Complete	173	19.05		56	18.01	117	19.60	0.238
HE Incomplete	23	2.53		2	0.64	21	3.52	0.087
HE Complete	11	1.21		3	0.96	8	1.34	0.634
Occupation								
Farmer	33	3.63		3	0.96	30	5.03	0.014
Autonomous	147	16.19		27	8.68	120	20.10	<0.001
Cook	274	30.18	<0.001	76	24.44	198	33.17	0.082
Gold miner	389	42.84		192	61.74	197	33.00	<0.001
Sex professional	16	1.76		5	1.61	11	1.84	0.916
Health professional	20	2.20		3	0.96	17	2.85	0.917
Other	29	3.19		5	1.61	24	4.02	0.644
Origin								
Pará	857	94.39		281	90.35	576	96.48	<0.003
Maranhão	26	2.86		14	4.50	12	2.01	0.540
Amazonas	6	0.66		5	1.61	1	0.17	0.408
Mato Grosso	6	0.66		3	0.96	3	0.50	0.051
Roraima	4	0.44		4	1.29	0	0	0.264
Amapá	2	0.22	<0.001	2	0.64	0	0	0.388
Goiás	2	0.22		0	0	2	0.34	0.769
Distrito Federal	1	0.11		0	0	1	0.17	0.464
Minas Gerais	1	0.11		0	0	1	0.17	0.464
Rondônia	1	0.11		1	0.32	0	0	0.464
Suriname	1	0.11		1	0.32	0	0	0.464
Venezuela	1	0.11		0	0	1	0.17	0.464

ES: Elementary School, HS: High School, HE: Higher Education. ^#^ One salary in Brazil: BRL 1,100.00/Month, approximately USD211.639 USD/Month * Chi-square grip test, ** Chi-square independence test (contingency table).

**Table 2 ijerph-18-10384-t002:** Odds ratio for malaria positivity in individuals from a mining region in the Brazilian Amazon.

Variables	*p*-Value	Odds Ratio	95% C.I. for OR	
			Lower	Upper
Gold miner	<0.001	4.387	2.744	7.012
>4 salaries	<0.001	1.950	1.301	2.922
2–3 salaries	<0.000	1.928	1.355	2.745
≤1 salary	0.353	0.772	0.446	1.334
Male	0.010	0.534	0.331	0.859

**Table 3 ijerph-18-10384-t003:** Epidemiological characteristics of malaria-positive individuals from a mining area in the Brazilian Amazon.

				N (Total)	%	*p*-Value
Etiological agent						
*P. vivax*				281	90.35	
*P. falciparum*				24	7.72	<0.001
*P. vivax + P. falciparum*				6	1.93	
Qualitative parasitemia (# of crosses)						
	*P. vivax*	*P. falciparum*	*P. vivax + P. falciparum*			
<0.5	36	5	0	41	13.18	
0.5	11	3	0	14	4.50	
1	27	2	0	29	9.32	
2	184	12	1	197	63.34	
3	23	2	0	25	8.04	
4	0	0	1	1	0.32	
2v–1f	0	0	1	1	0.32	<0.001
1v–0.5f	0	0	1	1	0.32	
2v–0.5f	0	0	1	1	0.32	
0.5v–2f	0	0	1	1	0.32	
Quantitative parasitemia						
<200				41	13.18	
200–300				14	4.50	
301–500				29	9.32	
501–10,000				197	63.34	<0.001
10,001–100,000				26	8.36	
>100,000				0	0.00	
500v–250f				1	0.32	
300v–10,000f				1	0.32	
5.000v–400f				1	0.32	
10.000v–250f				1	0.32	
Syphilis rapid test						
Positive				41	13.18	<0.001
Negative				270	86.82	
Syphilis VDRL						
Positive				29	9.32	
Negative				173	55.63	<0.001
Unrealized				109	35.05	
STIs in the last 12 Months						<0.001
Yes				25	8.04	
No				286	91.96	
How you treated						
Self-medication				9	36.00	0.3263
Did not treat				11	44.00	
Health service				5	20.00	
Tested for HIV						
Yes				146	46.95	0.3074
No				165	53.05	
Positive serum confirmation						
Yes				0	0.00	
No				142	97.26	<0.001
Uninformed				4	2.74	

**Table 4 ijerph-18-10384-t004:** Behavioral characteristics of malaria-positive individuals from a mining area in the Brazilian Amazon.

Variables	N (Total)	%	*p*-Value
Use of drugs in the last 12 months (licit and illicit)			
Yes	243	78.14	<0.001
No	68	21.86	
Type of drug			
Alcohol	230	94.65	
Cigarettes (smoking)	69	28.4	
Marijuana/Weed	33	13.58	<0.001
Aspirated Cocaine	11	4.53	
Crack	6	2.47	
Frequency of use of licit drugs			
Once a month	38	12.22	
Twice a month	2	0.64	
Once a week	41	13.18	
Twice a week	97	31.19	<0.001
3 times a week	1	0.32	
Every day	59	18.97	
None reported	73	23.47	
Frequency of use of illicit drug			
Once a month	8	2.57	
Twice a month	0	0	<0.001
Once a week	10	3.22	
Twice a week	14	4.5	
3 times a week	1	0.32	
Every day	11	3.54	
None reported	267	85.85	
Partner types			
Man	93	29.9	
Woman	208	66.88	
Man and Woman	1	0.32	<0.001
Not informed	9	2.89	
Number of partners in the last 30 days			
1	164	52.73	
2–8	108	34.73	<0.001
>8	30	9.65	
Not informed	9	2.89	
Gender identity			
Man	213	68.49	
Woman	98	31.51	<0.001
Sexual orientation			
Heterosexual	298	95.82	
Homosexual	3	0.96	<0.001
Bisexual	1	0.32	
Not informed	9	2.89	

## Data Availability

All relevant data is presented within the manuscript.

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
