# Peer review of "Socio-Epidemiological Features and Spatial Distribution of Malaria in an Area under Mining Activity in the Brazilian Amazon Region"

_ijerph, 2021, doi:10.3390/ijerph181910384_

Round 1

Reviewer 1 Report

The authors present interesting paper of demographic and epidemiological data from Brazil region Pará. Authors evaluated 908 probands from 101247 people from the region, which mean bellow 1% of population only. The dominance of male population is not surprising due to mining facilities in this region. From all patients only 311 have positive tests for malaria and some of them were even out region Pará. The presented paper had the standard segmentation, statistical evaluation is sufficient. Despite to interesting demographic results from the region of Pará, the manuscript needs corrections.

Here is mine comments and suggestions.

  1. The title of presented paper suggested the data relevant for malaria cases, but I do not know how the sexual transmitted diseases or drug abuse was related to malaria. Much more relevant information associated with malaria will be for example about water sources in the mining regions, used prophylaxes or anti-malaria counter measures which were not discussed in text. Please enhance the text about this information.
  2. Both figures were not in English and it should be difficult to fully understand of these figures.
  3. For the better awareness also the data from whole Brazil (or other part of Brazil) should be discussed, because I, as the reader, had no knowledge if the data from this region were different from other parts of Brazil or if the data were in agreement with the whole country.

Reviewer 2 Report

Dear Authors,
I don't find the presented article suitable for publishing. English needs substantial editing. I have struggled to follow the text, and figures are not translated at all. Still, the major drawback is the lack of any meaningful discussion of the obtained results.

Reviewer 3 Report

The premise of this paper is pretty good, that is that there are risk factors such as sex, occupation, income and geographic location that are linked to malaria transmission.  None of that is really new, though the location of the study is of interest.   The biggest problem with the manuscript is that the statistics are incomplete.  The authors note that study was a cross-sectional study in which malaria positivity of hospitalized patients was linked to potenital risk factors identified in a questionaire.  There are two issues with this: (1) the patients were all hospitalized and (2) the only statistic reported is a 'p' value.    The first issue is a matter of interpretation.   Hospitalized patients are probably different from the non-hospitalized population.  They may be more likely to live near the hospital, or may be wealthier.   This is an issue that should be addressed.   The statistics issue is a little more involved.  If this is a cross-sectional study, then the proper statistic to report should be a relative risk with a confidence interval (perhaps an odds ratio as an estimate of the relative risk with CI).   Without the relative risk, it is impossible to estimate the increase in risk.   The 'p' value indicates that there is a difference but does not indicate the direction of the difference.   There could be an increase in risk as a result of the risk factor or a decrease.   The 'p' just says there is a difference.  You can run logistic regression on the same data and get the OR, which would vastly improve the information you have reported here.   With only a 'p' value, the paper is probably not acceptable.

One other issue:  the use of "minimum wages" as a measure of wealth needs explantion.   Also, in table "lace range"?

Round 2

Reviewer 1 Report

The authors present interesting paper of demographic and epidemiological data from Brazil region Pará. Authors evaluated 908 probands from 101247 people from the region. Authors corrected paper according to reviews, text are more understandable and transparent. The manuscript should be accepted.